# A SOX2 Reporter System Identifies Gastric Cancer Stem-Like Cells Sensitive to Monensin

**DOI:** 10.3390/cancers12020495

**Published:** 2020-02-20

**Authors:** Diana Pádua, Rita Barros, Ana Luísa Amaral, Patrícia Mesquita, Ana Filipa Freire, Mafalda Sousa, André Filipe Maia, Inês Caiado, Hugo Fernandes, António Pombinho, Carlos Filipe Pereira, Raquel Almeida

**Affiliations:** 1i3S—Institute for Research and Innovation in Health, University of Porto, 4200-135 Porto, Portugal; dpadua@ipatimup.pt (D.P.); rbarros@ipatimup.pt (R.B.); aamaral@ipatimup.pt (A.L.A.); pmesquita@ipatimup.pt (P.M.); filipa.araujofreire@gmail.com (A.F.F.); mafsousa@ibmc.up.pt (M.S.); andre.maia@i3s.up.pt (A.F.M.); antonio.pombinho@ibmc.up.pt (A.P.); 2IPATIMUP—Institute of Molecular Pathology and Immunology, University of Porto, 4200-465 Porto, Portugal; 3Faculty of Medicine, University of Porto, 4200-319 Porto, Portugal; 4IBMC—Institute of Molecular and Cell Biology, University of Porto, 4200-135 Porto, Portugal; 5CNC—Center for Neuroscience and Cell Biology, University of Coimbra, 3004-517 Coimbra, Portugal; inesncaiado@gmail.com (I.C.); hugo.fernandes@uc.pt (H.F.); filipe.pereira@cnc.uc.pt (C.F.P.); 6Cell Reprogramming in Hematopoiesis and Immunity laboratory, Molecular Medicine and Gene Therapy, Lund Stem Cell Center, Lund University, BMC A12, 221 84 Lund, Sweden; 7Wallenberg Center for Molecular Medicine, Lund University, 221 84 Lund, Sweden; 8Faculty of Medicine, University of Coimbra, 3000-354 Coimbra, Portugal; 9Biology Department, Faculty of Sciences, University of Porto, 4169-007 Porto, Portugal

**Keywords:** cancer stem cells, gastric cancer, SOX2, monensin, SORE6-GFP reporter system, drug resistance, high-throughput screening

## Abstract

Gastric cancer remains a serious health burden with few therapeutic options. Therefore, the recognition of cancer stem cells (CSCs) as seeds of the tumorigenic process makes them a prime therapeutic target. Knowing that the transcription factors SOX2 and OCT4 promote stemness, our approach was to isolate stem-like cells in human gastric cancer cell lines using a traceable reporter system based on SOX2/OCT4 activity (SORE6-GFP). Cells transduced with the SORE6-GFP reporter system were sorted into SORE6+ and SORE6– cell populations, and their biological behavior characterized. SORE6+ cells were enriched for SOX2 and exhibited CSC features, including a greater ability to proliferate and form gastrospheres in non-adherent conditions, a larger in vivo tumor initiating capability, and increased resistance to 5-fluorouracil (5-FU) treatment. The overexpression and knockdown of SOX2 revealed a crucial role of SOX2 in cell proliferation and drug resistance. By combining the reporter system with a high-throughput screening of pharmacologically active small molecules we identified monensin, an ionophore antibiotic, displaying selective toxicity to SORE6+ cells. The ability of SORE6-GFP reporter system to recognize cancer stem-like cells facilitates our understanding of gastric CSC biology and serves as a platform for the identification of powerful therapeutics for targeting gastric CSCs.

## 1. Introduction

Gastric cancer (GC) is the fifth most common cancer and the third leading cause of cancer mortality worldwide [1,2,3]. It is a highly heterogeneous disease displaying various histopathological appearances and molecular signatures. The TNM classification remains the major prognostic marker used to inform treatment decisions in the clinic [4,5,6]. However, GC is highly heterogeneous even within the same TNM stage, and patient prognosis remains poor.

Cancer stem cells (CSCs) are cells that are capable of initiating and sustaining tumorigenesis and give rise to the phenotypically diverse and more differentiated progeny that composes the bulk of the tumor [7,8,9]. CSCs are thought to play a major role in driving disease recurrence, due to the increased therapeutic resistance caused by multiple molecular mechanisms, namely high expression of multidrug transporters, enhanced DNA repair, and altered cell-cycle kinetics [10,11,12,13]. Thus, understanding CSC biology will be crucial to the development of more effective cancer therapies.

Evidence of CSC existence has been found in a variety of malignancies, including GC [14,15,16,17,18,19]. Gastric CSCs have been identified in different studies with cell surface molecules namely CD44, CD133, ALDH1, and CD49f [19,20,21,22]. GC cells expressing these markers retain sphere-forming and tumor-initiating activities and some of them, namely CD44 and CD133, also have clinical relevance as markers of poor prognosis [23,24]. Thus, in GC, the identification of CSCs has been largely dependent on the combination of distinct cell surface markers that often differ among tumor types, and whose robustness in distinguishing tumorigenic from non-tumorigenic cells has been increasingly challenged [25,26,27]. Alternatively, transcription factors involved in the establishment of stemness features such as SOX2 and OCT4 could constitute more robust CSC markers [28].

The transcription factor SOX2 belongs to the group B of SOX proteins and plays an important role in cell fate determination, pluripotency, and cell differentiation during development in induced pluripotent stem cells (iPSCs) and in adult stem cells in multiple tissues [29,30,31,32,33]. Hence, SOX2 is unanimously recognized as a key pluripotency factor in multiple tissues and developmental stages. Importantly, in cancer, SOX2 has been increasingly associated with a CSC phenotype in tumors from different tissues, namely of the ovary, prostate, lung, skin, and central nervous system [34,35,36,37]. At the clinical level, SOX2 has been shown to act as an oncogene in several epithelial cancers and is a robust marker of poor prognosis [34]. In GC the role of SOX2 is less clear. We and others have shown that SOX2 associates with aggressive features, poor patient outcome, and worse response to chemotherapy [38,39,40,41]. However, other studies have shown the contrary, indicating that the relevance of SOX2 in gastric cancer needs further clarification, namely its association with the CSC phenotype [41,42].

OCT4 belongs to the family of POU-domain transcription factors and has a key role in pluripotency and somatic cell reprogramming [43]. Absence of or low OCT4 expression can lead murine embryonic stem cells to differentiate towards trophoblasts, while high expression of OCT4 can promote the differentiation of murine embryonic stem cells into primitive endodermal and mesodermal lineages. Additionally, neural differentiation of embryonic stem cells is enhanced by OCT4 overexpression [44,45]. The role of OCT4 in human cancers is much less studied than SOX2. It has been suggested as a CSC marker in some tumors such as gliomas, and squamous cell carcinomas of the lip and tongue, but the expression and relevance in GC has not been clarified [46,47,48,49].

In 2015, Tang et al. generated the novel SOX2-OCT4 reporter system designated the SOX2/OCT4 response element (SORE6-GFP) which they used to isolate breast cancer stem cells with self-renewal and tumor-initiating ability as well as drug resistance, paving the way to use this reporter system to identify gastric CSCs [50].

Thus, the main aims of this work were to isolate and characterize gastric CSCs from gastric carcinoma cell lines using the traceable reporter system based on SOX2/OCT4 activity. We further assessed the role of SOX2 in gastric CSC properties and searched for new drugs with increased activity against these cells.

## 2. Results

### 2.1. Isolation of a Small Cellular Subpopulation from Gastric Cancer Cell Lines Enriched in SOX2

AGS and Kato III cells transduced with SORE6-GFP were sorted into two subpopulations, SORE6+ and SORE6− cell populations, according to GFP expression. After cell sorting, in vitro and in vivo assays were used to assess CSC properties. An expression screening was performed to explore the mechanism behind the cellular response to chemotherapy and a high-throughput screening of pharmacologically active small molecules was done in order to identify drugs that can affect gastric CSCs (Figure 1).

We first showed that two human GC cell lines (AGS and Kato III) heterogeneously express SOX2. Both cell lines also express OCT4, although at low levels as assessed by qRT-PCR, and undetectable at the protein level by Western blot (Figure 2a). For this reason, we focused the study on SOX2. To characterize the properties of GC cells expressing SOX2, we used the SORE6-GFP reporter system, developed by Tang and his collaborators [50]. This reporter system enables the identification of cellular subpopulations enriched and depleted of these transcription factors. AGS and Kato III cells were transduced with the SORE6-GFP lentiviral system and selected with puromycin to ensure the presence of the reporter construct in all cells. Quantification of the cells that expressed GFP showed that they varied from 8.9 to 10.5% in AGS and from 8.0 to 16.6% in Kato III (Figure 2b). SORE6+ and SORE6− subpopulations from both cell lines were recovered by fluorescence-activated cell sorting (FACS) (Figure 2c) and analyzed for expression of the stem cell transcription factor SOX2 when reached approximately 80% confluence. SORE6+ cells showed an enrichment in SOX2 expression when compared with SORE6− cells and with the respective parental (wild-type, wt) cell lines (Figure 2d). In Kato III, SORE6− cells still expressed some level of SOX2, although lower than SORE6+ and the wt cell line. These results showed that the SORE6-GFP reporter system was able to identify a sub-population of cells, in GC cell lines, that was enriched in the stem cell transcription factor SOX2. Next, we evaluated the expression of cell-surface markers that have been previously identified as CSC markers in GC cell lines, namely ALDH1, CD44, CD133, and CD49f. We only observed an enrichment of ALDH1 in Kato III SORE6+ cells compared with SORE6− cells; however this was not the case in AGS cells where ALDH1 was not expressed. Interestingly, none of the other markers were enriched in the SORE6+ subpopulation from both cell lines (Figure 2e), supporting the idea that stemness transcription factors are better markers of CSCs. We further observed increased expression of the SOX2 targets, c-MYC and NOTCH1, in SORE6+ cells (Appendix A), as previously described [51].

### 2.2. SORE6+ Cell Population Recapitulates the Heterogeneous Cell Line

We then sought to compare the behavior of SORE6+ and SORE6− cellular subpopulations regarding CSC features. SORE6+ and SORE6− cells were kept in puromycin-containing medium after cell sorting to ensure the presence of the reporter construct. In both cell lines, GFP expression decreased with time in the SORE6+ subpopulation, whereas SORE6− cells could not give rise to GFP+ cells up to 10 weeks after sorting (Appendix A). AGS SORE6+ and SORE6− cell subpopulations were followed by time-lapse microscopy to monitor the emergence of cell heterogeneity. We observed the emergence of SORE6− cells in the SORE6+ subpopulation, reflecting the re-establishment of the heterogeneous population regarding SOX2 expression.

### 2.3. The SORE6+ Cells Were Highly Proliferative and Enriched in Tumor-Initiating Cells

We evaluated the proliferation of both sorted cellular subpopulations from AGS and Kato III through bromodeoxyuridine (BrdU) incorporation. SORE6+ cells had a significantly higher rate of proliferation when compared with the SORE6− cells and the corresponding wt cell lines (Figure 3a and Appendix A). A key property of CSCs is the ability to form sphere-like structures under non-adherent conditions and serum-free medium supplemented with growth factors to minimize the influence of external signals. In these conditions, the SORE6+ subpopulations from both cell lines were enriched in cells with increased ability to form gastrospheres, presenting a significantly higher number of gastrospheres larger than 100 µm in comparison to wt and SORE6− cells (Figure 3b). Then, we evaluated the tumor-initiating ability of SORE6+ and SORE6− cells after subcutaneous inoculation in athymic nude mice. The SORE6+ subpopulations were enriched in cells with improved tumor-initiating ability, since from 13 mice inoculated with AGS SORE6+ cells, 12 developed tumors and all mice inoculated with KATO III SORE6+ formed tumors. In contrast, only seven out of 13 mice inoculated with AGS SORE6− cells generated tumors and half of the 10 mice inoculated with Kato III SORE6− had a tumor (Figure 3c and Appendix A).

Tumors obtained from SORE6+ cells expressed SOX2 abundantly and some expression of SOX2 was also observed in tumors obtained from the SORE6− subpopulations. This was expected in KATOIII cell line, as SOX2 was never completely absent in the SORE6− cells. In contrast, in AGS SORE6− cells, SOX2 expression was completely absent in vitro but regained to some extent in vivo (Appendix A), suggesting phenotypic plasticity as previously described by Tang et al. [47].

### 2.4. The SORE6+ Cells Are More Resistant to 5-Fluorouracil (5-FU) Treatment

CSCs are more resistant to chemotherapeutic drugs, which is a critical property leading to tumor recurrence and significant clinical implications. To assess this property in the subpopulations obtained with the SORE6-GFP reporter system, cells were incubated with 5-FU, which is the standard-of-care in the treatment of GC, and the level of apoptosis was determined [52]. SORE6+ subpopulations from both cell lines were more resistant to 5-FU in comparison to the SORE6− cells and respective wt cell lines. In contrast, SORE6− subpopulations were the most sensitive to the drug. After 48 h of treatment with 5-FU, around 13% of AGS SORE6+ cells and about 55% of Kato III SORE6+ cells were in apoptosis. Conversely, approximately 77% of AGS SORE6− cells and 79% of Kato III SORE6− cells were apoptotic. AGS SORE6+ cells were more resistant to 5-FU than Kato III SORE6+ cells (Figure 4a and Appendix A). Apoptosis was caspase-dependent in AGS but not in KatoIII (Appendix A), as previously described [53]. To search for the molecular mechanism involved in drug resistance in SORE6+ cells, we used the RT2 Profiler PCR Array Human Cancer Drug Resistance kit which allowed us to profile the expression of 84 genes involved in cellular responses to chemotherapy (Figure 4b). The screening identified nine genes with a significantly different expression between AGS SORE6+ and AGS SORE6− cells. Of these, three were upregulated in AGS SORE6+ cells: BAX, CLPTM1L, and CYP3A5, and six were downregulated in these cells: CDKN1B, ELK1, ERBB2, IGF1R, SOD1 and RARG (Figure 4c). We next performed qRT-PCR for BAX, CLPTM1L, CYP3A5, CDKN1B, SOD1, and RARG in the two subpopulations from both cell lines. We obtained the same results in AGS, whereas in KatoIII only the upregulated genes were confirmed (Appendix A). These results suggest that relevant mechanisms of drug metabolism might be involved in the resistance to 5-FU and indicate also the existence of cell type specific drug resistance mechanisms.

### 2.5. SOX2 Has a Prominent Role in Determining the CSC Phenotype of SORE6+ Cells

In order to assess the relative role of SOX2 in defining the CSC properties previously observed in SORE6+ and SORE6− populations, we downregulated SOX2 expression in SORE6+ cells from both cell lines. SOX2 knockdown in SORE6+ cells resulted in a significant decrease in cell proliferation compared to the CTRL siRNA and the apoptosis analysis after 48 h treatment with 5-FU revealed that cells with SOX2 downregulation presented a significantly higher sensitivity to the drug in comparison with the CTRL siRNAs. Upon SOX2 knockdown, 49% of the cells were apoptotic, whereas only 21% of the cells treated with the CTRL siRNAs were in apoptosis. In Kato III SORE6+ cells with SOX2 downregulation, 61% of the cells were apoptotic, contrasting with the 38% of apoptotic cells detected in the control (Figure 5a). Overall, these results suggest that SOX2 plays a key role in determining the properties exhibited by SORE6+ cells. To further confirm this, we overexpressed SOX2 in the SORE6− subpopulation from both cell lines. The proliferation results showed that SORE6− cells transiently transfected with SOX2 significantly increased proliferation when compared with the control cells. They also became more resistant to 5-FU, with 44% of the cells overexpressing SOX2 in apoptosis, in contrast to the 76% of cells observed in the control, in the AGS SORE6− subpopulation. In Kato III SORE6− subpopulation, nearly 45% of cells transiently transfected with SOX2 entered apoptosis, whereas in the control, around 67% of cells were in apoptosis (Figure 5b).

### 2.6. Monensin Selectively Kills SORE6+ Cells

In order to identify new drugs that could be more efficient in targeting SORE6+ cells, we screened 1200 compounds from the Prestwick chemical library in both AGS SORE6 subpopulations and evaluated the loss in cell number caused by each one in the SORE6+ and SORE6− subpopulations. Compounds inducing a decrease in the cell number higher than three times the standard deviation (3SD) of the mean of the negative control (DMSO) were considered hits. From these primary hits, 41 compounds displayed a selective effect towards either SORE6+ or SORE6− subpopulations and were picked for testing their activity in a dose–response assay. Six compounds were then selected for validation based on their selective reduction of cell number towards one of the subpopulations and/or on the decrease of GFP+ cells within the SORE6+ population. This stepwise approach allowed us to identify two compounds that impacted preferentially on AGS SORE6+ and four on AGS SORE6− cell number. One of these compounds, monensin, more efficiently decreased the cell number of the AGS SORE6+ cells (Figure 6a) while preferentially targeting the GFP+ cells within this population (Figure 6b). This result was replicated in Kato III SORE6 + and SORE6− cells where, although a statistically different effect on cell number could not be observed (Appendix A), the GFP+ subpopulation within the Kato III SORE6+ cells was significantly more affected by monensin than the GFP- subpopulation (Appendix A). The identification of monensin’s ability to reduce the number of cells with high GFP levels in the SORE+ population suggests that there is a possibility for CSCs in gastric tumors to be therapeutically targeted.

## 3. Discussion

In this study, making use of the SORE6-GFP reporter system, we identified two cellular subpopulations, SORE6+ and SORE6− cells, with differential expression of SOX2 in two GC cell lines. Further, we demonstrated that SORE6+ cells present features of CSCs mostly regulated by SOX2. We further identified differentially expressed genes that could be involved in drug resistance of SORE6+ cells and monensin as a new drug with superior toxicity in gastric CSCs.

The CSC phenotype has been extensively studied and it has become increasingly clear that one of the main challenges is the identification of reliable CSC markers. An essential tool for the identification and isolation of candidate CSCs has been the expression of unique combinations of cell surface markers, which is an approach that has allowed the isolation of CSCs in many solid malignancies using flow cytometry or magnetic cell sorting. In GC, a number of cell surface markers have been identified as CSC markers (CD44, CD133, ALDH1, and CD49f) [19,21,22,54]. However, several studies revealed that these markers lack specificity and furthermore cannot be used for real-time assessment of the CSC behavior [26,55]. To address this issue, Tang et al. developed a new lentiviral reporter system (SORE6-GFP) based on SOX2/OCT4 stemness marker activity and successfully used it to identify, quantify and isolate a small subpopulation with CSC characteristics in breast cancer cell lines and in the prostate [50,56]. Here, it was used for the first time in GC cell lines and showed that SOX2 is not only a marker but also a driver of the CSC phenotype in GC. SOX2 is an essential component in the cocktail of transcription factors used to generate iPSCs. It is required in embryonic stem cells and it is expressed in adult multipotent stem cells in different tissues such as the stomach, esophagus, lung, skin, and brain [31]. Thus, it is conceivable that it might be a marker and an inducer of the CSC phenotype. Sox2-expressing cells are able to differentiate into all cell types found in a gastric unit, and ablation of the Sox2+ population prevents renewal of the gastric epithelium [31]. SOX2 expression has been associated with multiple tumor types; however, at the clinical level there is a lack of consensus as to whether it is a marker of good or poor prognosis. Still, the vast majority of studies associate SOX2 expression with poor patient prognosis [37,39]. We have previously shown that SOX2 is a marker of poor patient prognosis in GC, which would be in accordance with the more aggressive behavior of the cells that express SOX2, namely increased proliferation, increased tumor-formation, and increased drug resistance, observed in this study [38]. In addition, it is well established that SOX2 regulates cellular proliferation in many cancer types; for instance, its overexpression in the lungs of mice induces rapid proliferation and, in some cases, leads to adenocarcinomas [57]. On the other hand, the knockdown of SOX2 in different cell lines, including gastric, significantly reduces proliferation in vitro and tumor growth [40,58] SOX2 has also been shown to promote tumor invasion, migration, and metastasis in numerous cancers [59]. In cervical squamous cell carcinoma, high expression of SOX2 along with high expression of OCT4 indicated resistance to radiotherapy, and both factors were important predictors of poor patient survival [60]. Our results are in accordance with these studies and with the recent study by Chen et al. that shows that OCT4 and SOX2 overexpression in GC cells induces changes in the biological behavior of the cells towards a CSC phenotype [61]. However, in our study, the reporter system used allowed us to separate cells based on the endogenous expression of these transcription factors, mainly SOX2. Furthermore, we observed concordance between SOX2 and ALDH1A1 expression in Kato III, but not with the other previously identified gastric CSC-surface markers. This might result from the existence of multiple CSC populations or CSC plasticity, which requires further studies [62]. In this study, we highlight the relevance of SOX2 in the behavior of gastric cancer but there are other molecular players, such as the recently identified USF1 whose interaction with the CSC phenotype needs to be addressed [63].

An important clinical readout of the CSC phenotype is the increased drug resistance, namely to 5-FU in GC, which in patients might be related with the worse prognosis, already described [64,65]. We confirmed that SORE6+ cells have increased resistance to apoptosis and screened genes that could be involved in this process. Interestingly we found that SORE6+ cells have increased expression of two genes known to enhance resistance to drugs by different mechanisms. CLPTM1L, also known as cisplatin resistance-related protein 9, confers resistance to apoptosis caused by genotoxic agents and leads to up-regulation of the anti-apoptotic protein, Bcl-xL [66,67]. CYP3A5 encodes a member of the cytochrome P450 superfamily of enzymes, which have multiple functions in drug metabolism and in facilitating the elimination of drugs [68]. Interestingly, SORE6− cells were enriched in, among other entities with a less clear role, a known inhibitor of proliferation (CDKN1B) and an inducer of differentiation (RARG), which could be contributing to the less aggressive phenotype of these cells [69,70]. These results indicate that the mechanisms involved in drug resistance in the SORE6+ cells, are not related with regulation of the cell cycle. These cells are more proliferative, which in principle should render them more sensitive to 5-FU; however, this is not the case. Here, we obtained clues that multidrug resistance mechanisms are at stake in the SORE6+ cells, in accordance to what has been demonstrated for CSCs in multiple tumors [65,71,72].

The reporter system used in this study proved to be useful for the screening of drugs with impact in CSCs. We identified monensin, a polyether ionophore antibiotic, currently used as an anti-bacterial and anti-parasitic drug, to be toxic to cancer cells with superior activity over CSCs [73]. Ionophores are organic molecules that bind metal cations and form lipid-soluble complexes, facilitating their transport across cellular membranes [74]. Like other ionophore antibiotics, monensin induces apoptosis and causes growth inhibition in diverse types of cancer cells, even in those that display multidrug resistance [75,76]. At the molecular level, monensin has pleiotropic effects affecting multiple cancer-related pathways such as the E2F/DP1, STAT1/2, NFκB, AP-1, Wnt, and Elk-1/SRF pathways, likely dependent on the cellular model [77]. Interestingly, another structurally related compound, the ionophore salinomycin, was shown to specifically kill human CSCs and inhibit in vivo tumorigenesis. Salinomycin induces cell apoptosis, disturbing the equilibrium of Na+/K+ ions in cellular membranes including mitochondria and cytoplasm, overcoming resistance to drugs by multidrug-resistant genes [78]. In this study, we showed that SORE6+ cells were resistant to 5-FU, but sensitive to monensin, suggesting that monensin is able to bypass the common mechanisms of drug resistance present in CSCs. Being an ionophore, monensin also alters the concentration of cations inside the cells. This has been demonstrated in prostate cancer cell lines, where intracellular Ca+ levels were reduced in the presence of the drug and this was the primary mechanism leading to cell cycle arrest and apoptosis [79].

## 4. Materials and Methods

### 4.1. Cell Culture

Two human gastric carcinoma cell lines were used in this study, AGS (ATCC, CRL-1739, Manassas, VA, USA) and Kato III (ATCC, HTB-103, Manassas, VA, USA), as well as the human embryonic kidney cell line, HEK293T (kindly provided by Dr. João Relvas from Glial Cell Biology Group, i3S, Porto, Portugal). AGS and Kato III cell lines were cultured in RPMI medium 1640 with 25 mM Hepes and GlutaMAX-1 (Gibco, Life Technologies, Thermo Fisher, Waltham, MA, USA) supplemented with 10% fetal bovine serum (FBS) (Biowest, France). After sorting, the cell lines were cultured in DMEM without phenol red (Gibco, Life Technologies, Thermo Fisher, Waltham, MA, USA) supplemented with 10% FBS, 1% sodium pyruvate (Gibco, Life Technologies, Thermo Fisher, Waltham, MA, USA), and 1% penicillin-streptomycin (Gibco, Life Technologies, Thermo Fisher, Waltham, MA, USA). The HEK293T cell line was cultured in DMEM with 4.5 g/L D-glucose, L-glutamine, and pyruvate (Gibco, Life Technologies, Thermo Fisher, Waltham, MA, USA). All cell lines were maintained at 37 °C in a humidified atmosphere containing 5% CO_2_.

### 4.2. Viral Supernatant Production and Cell Transduction

The mCMVp-dsCopGFP-PURO (Ø) or SORE6-mCMVp-dsCopGFP-PURO (SORE6-GFP) lentiviral reporters were a kind gift from Dr. Lalage M. Wakefield, NCI, USA. The SORE6-GFP is a lentiviral reporter construct in which six concatenated repeats of a composite SOX2/OCT4 response element (SORE6) from the proximal human NANOG promoter were coupled to a minimal cytomegalovirus (CMV) promoter and used to drive expression of GFP [50]. SORE6-GFP and the empty vector were cotransfected with the psPAX2 packaging plasmid and the pVSV-G envelope plasmid into HEK293T cells using Lipofectamine 2000 (Invitrogen, Carlsbad, CA, USA) in a 1 µg DNA: 1.5 µL lipofectamine ratio in serum- and antibiotic-free Opti-MEM medium (Gibco, Life Technologies, Thermo Fisher, Waltham, MA, USA), and cells were incubated for 24 h. After incubation, cells were exposed to fresh HEK293T culture medium and left undisturbed for 48 h. All cell lines were exposed to viral supernatants (Ø or SORE6-GFP) diluted 1:5 in fresh medium for 72 h. Transduced AGS and Kato III cultures were selected with 5 μg/mL of puromycin (Gibco, Life Technologies, Thermo Fisher, Waltham, MA, USA) for at least 10 days prior to fluorescence-activated cell sorting (FACS), to ensure the presence of SORE6-GFP vector in all cells. GFP expression was monitored with the ZOE Fluorescent Cell Imager (Bio-Rad, Hercules, CA, USA).

### 4.3. Fluorescence-Activated Cell Sorting (FACS)

Transduced cells were collected by trypsinization, washed with phosphate-buffered saline (PBS), and resuspended in PBS with 5% FBS. AGS and Kato III cells transduced with SORE6-GFP were sorted using a FACS Aria II Cell Sorter flow cytometer (BD Bioscience, Franklin Lakes, NJ, USA), and cells transduced with Ø viral supernatant were used as negative controls for gating assessment. Cells with and without GFP signals—SORE6+ and SORE6−, respectively—were seeded in six-well plates at a density of 3.5 × 10^4^ to 7.5 × 10^4^ cells/mL. The data were analyzed using FlowJo software (version 7.6.1).

### 4.4. Western Blot and Immunofluorescence

To prepare whole cell extracts, cell pellets were washed with PBS and resuspended in cold RIPA lysis buffer (50 mM Tris-HCl pH = 7.4, 150 mM NaCl, 2 mM EDTA, 1% NP-40, 0.1% SDS) supplemented with the complete protease inhibitor cocktail (Roche, Basel, Switzerland), 1 mM PMSF, and 1 mM Na_3_VO_4_. Then, 30 µg of total protein extracts were separated by standard SDS-PAGE using the Precision Plus Protein Standard Dual Color protein marker (Bio-Rad, Hercules, CA, USA). After separation, proteins were transferred to a nitrocellulose membrane (Amersham, GE Healthcare, Chicago, IL, USA). Membranes were washed with TBS-1% Tween-20 (Sigma-Aldrich, St. Louis, MO, USA) and blocked with either 5% non-fat milk in TBS-1% Tween-20 or 5% BSA (Sigma-Aldrich, St. Louis, MO, USA) for 1 h at room temperature. The membranes were incubated overnight at 4 °C with the following primary antibodies: SOX2 (diluted 1:500; Cell Marque, Sigma-Aldrich, St. Louis, MO, USA); CD49f (ITGA6) (diluted 1:1000; Sigma-Aldrich, St. Louis, MO, USA); CD133 (diluted 1:1000; Cell Signaling Technology, Danvers, MA, USA); CD44 (diluted 1:500; Cell Signaling Technology, Danvers, MA, USA); ALDH1A1 (diluted 1:500; Abcam, Cambridge, UK); c-Myc (diluted 1:1000; Cell Signaling Technology, Danvers, MA, USA); NOTCH1 (diluted 1:500; Santa Cruz Biotechnology, Dallas, TX, USA); PARP (diluted 1:1000; Cell Signaling Technology, Danvers, MA, USA); cleaved PARP (diluted 1:1000; Cell Signaling Technology, Danvers, MA, USA); caspase 3 (diluted 1:1000; Cell Signaling Technology, Danvers, MA, USA); cleaved caspase 3 (diluted 1:1000; Cell Signaling Technology, Danvers, MA, USA); caspase 7 (diluted 1:1000; Cell Signaling Technology, Danvers, MA, USA); caspase 9 (diluted 1:1000; Cell Signaling Technology, Danvers, MA, USA); and *β*-actin (diluted 1:4000; Santa Cruz Biotechnology, Dallas, TX, USA). Membranes were washed with TBS-1% Tween-20 and incubated for 1 h at room temperature with the respective HRP-conjugated secondary antibody: rabbit IgG (HRP) (diluted 1:2000; Santa Cruz Biotechnology, Dallas, TX, USA); mouse IgG (HRP) (diluted 1:2000; Santa Cruz Biotechnology, Dallas, TX, USA), or goat IgG (HRP) (diluted 1:2000; Santa Cruz Biotechnology, Dallas, TX, USA). Signal detection was performed using the ECL detection kit (Amersham, GE Healthcare, Chicago, IL, USA). *β*-actin was used as loading control. For immunofluorescence analysis, AGS and Kato III cell lines were plated in six-well plates, with five coverslips in each well, at a density of 6 × 10^4^ cells/mL in RPMI 10% FBS and allowed to attach overnight. When cells reached about 70%–80% confluence, medium was removed, cells were washed with PBS (1X), fixed with cold methanol and washed again with PBS (1X). Then, the coverslips were blocked with 1 µL goat normal serum (Dako, Denmark): 5 µL PBS 0.5% Tween-20 0.05% BSA for 30 min in the dark at room temperature. Next, the coverslips were incubated with the primary antibody SOX2 (diluted 1:50) overnight at 4 °C. After incubations, the coverslips were washed twice with PBS (1X) and incubated with the secondary antibody goat anti-rabbit Alexa Fluor 488 (diluted 1:100; A11034, Thermo Fisher, Waltham, MA, USA) for 45 min in the dark at room temperature. 4′,6-diamidino-2-phenylindole (DAPI, dilution 1:100; Sigma-Aldrich, St. Louis, MO, USA) was added to the coverslips for nuclei staining. The coverslips were mounted with Vectashield mounting medium (Vector Laboratories, Burlingame, CA, USA) and pictures were taken using a Carl Zeiss fluorescent microscope (Carl Zeiss Microscopy, Oberkochen, Germany). Uncropped western blots are shown in Appendix A.

### 4.5. Quantitative Real-Time PCR (qRT-PCR)

Total RNA was extracted using the TRI Reagent (Sigma-Aldrich, St. Louis, MO, USA), according to the manufacturer’s protocol. Concentration and quality of the RNA were assessed using the NanoDrop ND-1000 spectrometer (V3.5.2 Software). Total RNA was converted to cDNA for 10 min at 65 °C in a thermocycler (Bio-Rad, Hercules, CA, USA) in a final reaction volume of 20 µL, containing 1 µL of random primers (100 ng/µL), 4 µL of Buffer 5x, 2 µL of DTT, 2 µL of dNTPs (10 mM), 0.3 µL of RNAseOUT (40 U/µL), and 0.5 µL of SuperScript III Reverse Transcriptase (200 U/µL) in DEPC-treated water (all reagents were purchased from Invitrogen, Carlsbad, CA, USA). For the PCR, each reaction was prepared with 4 µL of cDNA diluted 1:20 in DEPC-treated water, 10 µL Power SYBR Green PCR Master Mix (Applied Biosystems, Foster City, CA, USA), 0.6 µL of each primer (10 µM), and 4.8 µL of DEPC-treated water (Invitrogen, Carlsbad, CA, USA). The primers used were the following: 18SFor: CGCCGCTAGAGGTGAAATTC; 18SRev: CATTCTTGGCAAATGCTTTCG; OCT4For: TGCAGCAGATCAGCCACAT; OCT4Rev: ACACTGGTCCCCCTGAGAAA; BAXFor: TTTTGCTTCAGGGTTTCATC; BAXRev: GACACTCGCTCAGCTTCTTG; CLPTM1LFor: TCTGATACACAGCAGATCGAGG; CLPTM1LRev: AGTTGTCCGCCATCACGTTC; CYP3A5For: TCGAAGGTCTTTAGGCCCAG; CYP3A5Rev: GGTGAAGGTTGGAGACAGCA; CDKN1BFor: CCAAAGGTGCCTGCAAGGTG; CDKN1BRev: AGAAGAATCGTCGGTTGCAGGT; RARGFor: CATCCAAGAGACTGCCCGAC: RARGRev: CGTAGCTGCTGGAGTGGG; SOD1For: GCCAAAGGATGAAGAGAGGCAT; SOD1Rev: ACATCGGCCACACCATCTTT. The reactions were performed in a 7500 Fast Real-Time PCR System using the software v2.0.6 (Applied Biosystems, Foster City, CA, USA). Each experiment was carried out in triplicates and three negative controls (without cDNA) were included in each plate. The housekeeping control gene, 18S, was used for normalization of target gene abundance. Data was analyzed by the comparative 2ΔΔCT method [80].

### 4.6. Sulforhodamine B (SRB) Assay

For the 5-fluorouracil (5-FU, Sigma-Aldrich, St. Louis, MO, USA) IC50 determination, a dose–response curve between the drug concentration and the percentage of cell growth inhibition was performed for each cell line. For that, AGS and Kato III cell lines were seeded in 96-well plates at a density of 1 × 10^4^ and 2 × 10^4^ cells/mL respectively and treated for 48 h with a range of concentrations between 6.25 and 100 nmol/mL of 5-FU (Sigma-Aldrich, St. Louis, MO, USA). Cells were fixed with cold 10% (w/v) TCA (Merck Millipore, Burlington, MA, USA), stained with 0.4% (w/v) SRB (Sigma-Aldrich, St. Louis, MO, USA) for 30 min and then washed with 1% (w/v) acetic acid. The dye was solubilized with 10mM Tris Base (pH = 10.5) and the optical densities were measured at 510 nm using the Synergy Microplate Reader (Biotek, Winooski, VT, USA).

### 4.7. Annexin V/Propidium Iodide (PI) Assay

Cell lines were seeded in six-well plates at a density of 0.6 × 10^5^ cells/mL and were allowed to attach overnight. The following day, cells were treated with the concentration of 5-FU corresponding to the IC50 value determined for AGS and Kato III wild-type cell lines. Cells treated only with dimethyl sulfoxide (DMSO, Applichem) were used as control. After 48 h, cells were collected by trypsinization and pellets were resuspended in Binding Buffer 4x (annexin V Apoptosis detection Kit, eBioscience, Affymetrix, Santa Clara, CA, USA). Each sample was first stained with annexin V Alexa Fluor 594 conjugate (Invitrogen, Carlsbad, CA, USA) or annexin V APC (ImmunoTools, Germany) and then with propidium iodide (PI) (annexin V Apoptosis detection Kit, eBioscience, Affymetrix, Santa Clara, CA, USA). The percentage of apoptosis was measured using the FACS Aria II flow cytometer. The results were analyzed using FlowJo software (version 7.6.1).

### 4.8. BrdU Incorporation Assay

Cells were cultured in six-well plates at a density of 0.8 × 10^5^ cells/mL and, 24 h later BrdU was incorporated in the cell culture medium at the ratio of 1:1000, for 1 h to 2 h at 37 °C. The cells were harvested by trypsinization and cell pellets fixed in ice-cold methanol (Chem-Lab, Zedelgem, Belgium). The cells were resuspended in HCl 4 M (Mallinckrodt Baker, Avantor, Allentown, PA, USA) for 20 min at room temperature followed by two washing steps with PBS and a blocking step with PBS containing 0.5% Tween 20 and 0.05% BSA for 10 min. The cells were incubated with the primary antibody against BrdU (1:20, monoclonal mouse, Bu20a, Dako, Denmark) for 1 h at room temperature, washed with PBS, and incubated with the secondary antibody labelled with FITC (1:150 µL, polyclonal rabbit anti-mouse, Dako, Denmark) for 30 min, washed again and resuspended in PBS. The percentage of BrdU positive cells was assessed using the FACS Aria II flow cytometer. The data was analyzed using FlowJo software (version 7.6.1). BrdU incorporation was also observed by immunofluorescence, methodology previously described in 4.4, with an additional step after methanol fixation where the cells were treated with HCl 4 M for 20 min at room temperature. The coverslips were incubated with the primary antibody BrdU (diluted 1:20) overnight at 4 °C. The goat anti-mouse Alexa Fluor 594 (diluted 1:100; A11032, Thermo Fisher, Waltham, MA, USA) was used as secondary antibody.

### 4.9. Gastrosphere Formation

Cells were trypsinized and disaggregated with a 25-gauge needle (BD Bioscience; Franklin Lakes, NJ, USA) to obtain single-cell suspensions in cold PBS. Cells were then plated at 1 × 10^3^ cells/well in 24-well plates coated with 12 g PolyHEMA (Sigma-Aldrich, St. Louis, MO, USA)/1 L of 95% EtOH. Cells were grown for 10 days in serum-free DMEM medium containing B27 supplement, N2 supplement, 10 ng/mL of bFGF, 20 ng/mL of hEGF, and 1% of penicillin-streptomycin, at 37 °C and 5% CO_2_ in a humidified incubator. After 10 days, cells were examined and the number of gastrospheres with more than 100 μm in diameter were counted in a total of 10 independent fields per well in a Zeiss optical microscope.

### 4.10. Xenografts in Nude Mice

Animal experiments were approved by the i3S animal ethics committee (CEA–Comissão de Ética Animal) and licensed by the Direção Geral de Alimentação e Veterinária (DGAV), code DAGV 0421/000/000/2015. Experiments were performed in the animal facility of i3S according to the institutional guidelines regulated by Decreto-Lei 113/2013 (Portugal), which is the national transposition of the European Directive 2010/63/EU on the protection of animals used for scientific purposes and FELASA guidelines. Female athymic nude mice (NIH(S)II-nu/nu [81]) aged 6 to 8 weeks were subcutaneously inoculated with 5 × 10^5^ AGS SORE6+ or AGS SORE6− cells or with 3 × 10^5^ Kato III SORE6+ or Kato III SORE6− cells resuspended in 1:1 RPMI medium: Matrigel matrix phenol red-free (BD Bioscience; Franklin Lakes, NJ, USA) using a 25 gauge needle. Animals were observed, and their weight measured once a week. They were euthanized one to two months after inoculation and tumors were resected and fixed in neutral buffered formalin 10%.

### 4.11. Immunohistochemistry

After fixation, the mice tumors were processed, embedded in paraffin and sectioned with a microtome (Microm HM 335 E) at 3-µm thickness. Tumor sections were deparaffinized in xylene and rehydrated through a graded series of ethanol (EtOH) concentrations. Antigen retrieval was performed by heat-induced (98 °C) in 1 mL EDTA buffer 10 mM pH = 8.0 (Thermo Fisher, Waltham, MA, USA): 10 mL dH_2_O for 40 min followed by 20 min at room temperature. The sections were then washed with TBS-0.5% Tween-20 and incubated with hydrogen peroxide 10% (v/v). Primary antibody SOX2 was diluted 1:50 in antibody diluent OP Quanto (Thermo Fisher, Waltham, MA, USA) and incubated overnight at 4 °C in a humidified chamber. After incubation, slides were washed and incubated with the secondary antibody HRP polymer (Rabbit/Mouse Dako Real EnVision, Denmark) for 30 min at room temperature. Next, the slides were washed twice with TBS-0.5% Tween-20 and incubated with Substrate Buffer mix with DAB + Chromogen (50X) (Dako Real, Denmark), according to the manufacturer’s instructions, for 3 min at room temperature in a humidified chamber. Slides were then washed with water and counterstained with hematoxylin for 2 min and washed again. Finally, the slides were dehydrated in a series of EtOH dilutions, passed through xylene and mounted with permanent mounting medium (Bio-mount HM; Bio-Optica, Milan, Italy).

### 4.12. SOX2 Overexpression and Knockdown

Cells were seeded in six-well plates at a density of 0.6 × 10^5^ cells/mL and incubated for 24 h at 37 °C and 5% CO_2_ in a humidified incubator. For SOX2 overexpression, AGS and Kato III SORE6− cells were transiently transfected with a SOX2 expression vector or with the corresponding empty vector (pcDNA3.1) in a 1 µg DNA: 1.5 µL lipofectamine ratio in serum- and antibiotic-free Opti-MEM medium. For SOX2 knockdown, AGS SORE6+ and Kato III SORE6+ cells were transiently transfected with either a commercial set of three stealth siRNAs directed against human SOX2 (HSS144045, HSS186041, and HSS186042) or an siRNA negative control (CTRL siRNA), all from Invitrogen (Carlsbad, CA, USA). The transfection was performed using Lipofectamine 2000 with 100 nM of siRNAs targeting SOX2 or siRNA negative control (CTRL) in serum- and antibiotic-free Opti-MEM medium. After 24 h, the transfection medium was replaced with DMEM without phenol red supplemented with 10% FBS and 1% sodium pyruvate. Twenty-four hours later, the cells were submitted to annexin V/PI assay and BrdU incorporation assay. The efficiency of SOX2 overexpression and knockdown was verified by Western Blot and qRT-PCR respectively.

### 4.13. Real-Time2 Profiler PCR Array (RT2PCR)

We used the RT2 Profiler^TM^ PCR Array Human Cancer Drug Resistance (Qiagen, Hilden, Germany) to identify genes involved in drug resistance that could be differentially expressed between SORE6− and SORE6+ cells. AGS SORE6+ and SORE6− total RNA was extracted, quantified and mRNA converted to cDNA using the same procedures previously mentioned. For the PCR, a master mix was prepared, according to the manufacturer’s instructions, with 102 µL of a cDNA solution (91 µL of RNase-free water and 20 µL of cDNA), 1350 µL Power SYBR Green PCR Master Mix (Applied Biosystems, Foster City, CA, USA), and 1248 µL of RNase-free water (NZYTech, Lisboa, Portugal). Then, 19 µL of the mix were distributed in each well of a 96-well plate. The reactions were performed in a 7500 Fast Real-Time PCR System using the software v2.0.6 (Applied Biosystems, Foster City, CA, USA). The average of five housekeeping control genes included in the kit was used for normalization of target gene abundance. Data were analyzed by the comparative 2ΔΔCT method [80].

### 4.14. Small Molecules High-Throughput Screening

AGS SORE6+ and SORE6− cells were seeded in 384-well plates (Greiner CELLSTAR^®^ 781091) at a density of 5 × 10^4^ cells/mL in RPMI medium supplemented with 10% FBS and allowed to attach overnight. The next day, 1200 compounds from the Prestwick Chemical Library were transferred, one compound per well, using a pintool (V&P Scientific, San Diego, CA, USA) coupled to the MDT head of a JANUS Automated Workstation (PerkinElmer, Waltham, MA, USA) for a final concentration of 8 µM in DMSO. For each plate, four wells were treated with DMSO 0.4% and used as negative controls, and quadruplicate wells were treated with 5-FU (4.5 µg/mL) and used as positive controls. Cells were incubated for 48 h after which they were washed with PBS, fixed with 4% PFA (Alfa Aesar, Haverhill, MA, USA), and stained with Hoechst (1 µg/mL). The image acquisition was performed in an INCell Analyzer 2000 (GE Healthcare, Chicago, IL, USA) with a Nikon 10x/0.45 NA Plan Fluor objective. Four fields of view were acquired per well covering the whole well. Image analysis was executed using the INCell Investigator software (GE Healthcare, Chicago, IL, USA). Briefly, the image analysis workflow consists in the segmentation of the nuclei, from the Hoechst channel, followed by a slight expansion of the nuclear mask to cover a larger area of the cell. The mean pixel intensity of the GFP channel has been extracted for each individual cell. Spotfire software (TIBCO, Palo Alto, CA, USA) was used to visualize the data, perform quality control of the image segmentation and GFP+/GFP− threshold decision. A custom-made MATLAB script (version R2018, MathWorks, Natick, MA, USA) was developed to automatically extract specific and conditional data of all analyzed cells and per each condition. Hits were identified using thresholds for total cells and GFP positive cells loss higher than three times the standard deviation (3SD) of the mean of the negative controls and loss of GFP positive cells. Primary hit confirmation was performed testing each compound in a dose–response assay (0.125 µM–16 µM) in triplicate. For hit validation, compounds were repurchased and tested again in dose–response in cells were seeded in 96-well plates (CellCarrier-96 Ultra, PerkinElmer, Waltham, MA, USA) at a density of 2.5 × 10^4^ cells/mL and incubated for 24 h at 37 °C and 5% CO_2_ in a humidified incubator. Then, cells were exposed to the compounds for 48 h and then fixed, stained, and analyzed as previously described for the primary screening. DMSO was used as a negative control and 5-FU used as a positive control.

### 4.15. Statistical Analysis

Results were expressed as mean ± SD from at least three independent experiments. Statistical significance was determined by unpaired two-tailed t test or one-way ANOVA followed by post hoc Turkey’s test using GraphPad Prism 8.0 software (GraphPad Software, La Jolla, CA, USA). Differences were considered significant when *p* ≤ 0.05.

## 5. Conclusions

We demonstrated that SOX2 has a prominent role in gastric CSCs and can be used as a marker of these cells. Furthermore, we identified a new drug, monensin, with increased toxicity over gastric CSCs. We showed for the first time that monensin acts on CSCs, paving the way for the clinical testing of this drug in GC patients.

## Figures and Tables

**Figure 1 cancers-12-00495-f001:**
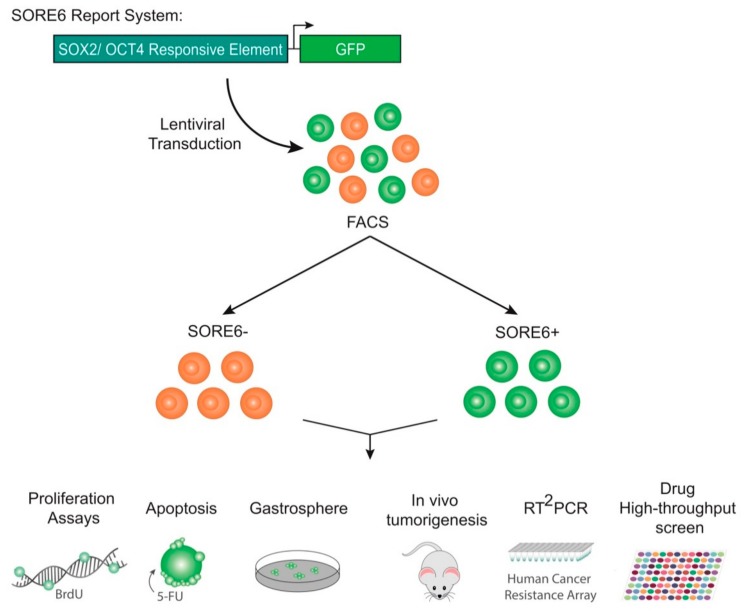
Scheme of overall approach showing the assays performed after the isolation of SORE6+ and SORE6− populations from gastric cancer cells transduced with the SORE6-GFP reporter system.

**Figure 2 cancers-12-00495-f002:**
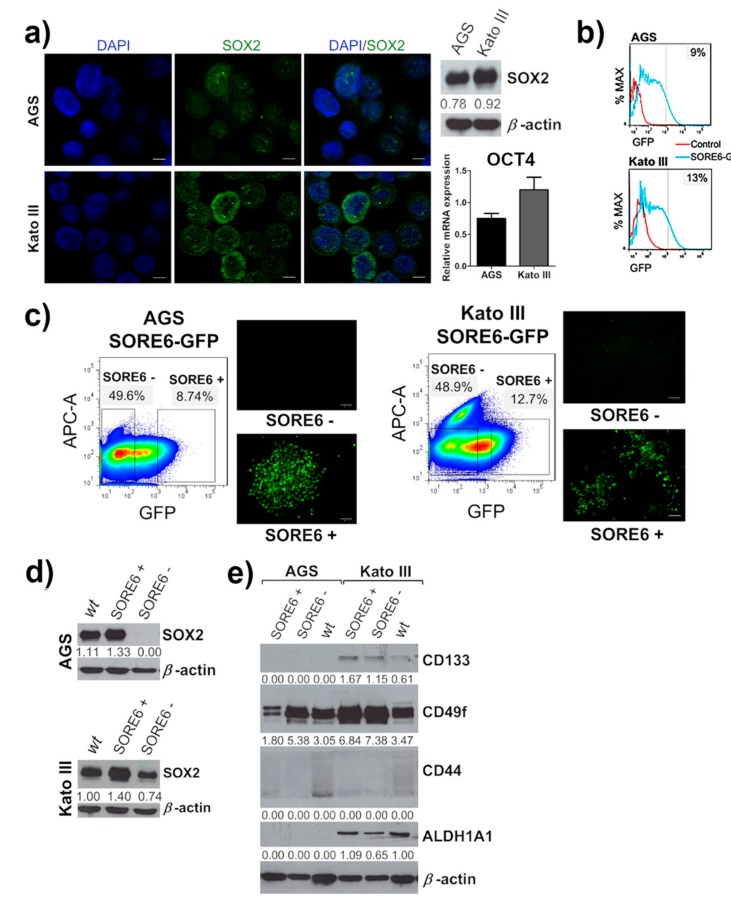
SORE6+ cells have a higher expression of SOX2 than SORE6− cells. (**a**) Western blot and qRT-PCR analysis exhibiting the expression of SOX2 and OCT4, respectively, in AGS and Kato III cell lines. The representative immunofluorescence images reveal the differences of SOX2 expression per cell in both cell lines. (**b**) Fluorescence-activated cell sorting (FACS) analysis of AGS and Kato III 10 days after transduction with SORE6-GFP reporter. (**c**) FACS analysis with gate selection and cell sorting of two cell subpopulations, SORE6+ cells (with GFP expression) and SORE6− cells (without GFP expression), in both SORE6-GFP cell lines, followed by fluorescence images of the respective populations sorted. Scale bar = 100 µm. APC-A corresponds to an empty channel used to exclude autofluorescent cells from the analysis. (**d**) SOX2 expression evaluated by Western blot in the FACS-sorted SORE6+ and SORE6− cells from both cell lines. (**e**) Western blot analysis of the putative gastric cancer stem cell markers ALDH1A1, CD44, CD49f, and CD133 on sorted AGS SORE6 and Kato III SORE6 cell populations. In Western blot *β*-actin was used as an internal control. Numbers below every Western blot picture represent a semi-quantitative analysis of each line (intensity ratio: “gene of interest”/*β*-actin). Pictures show cropped areas of Western blots, the whole images are included in the Appendix A. The qRT-PCR results were normalized to 18S expression. wt corresponds to the parental cell line (AGS or Kato III). Results are mean ± SD of at least three independent experiments.

**Figure 3 cancers-12-00495-f003:**
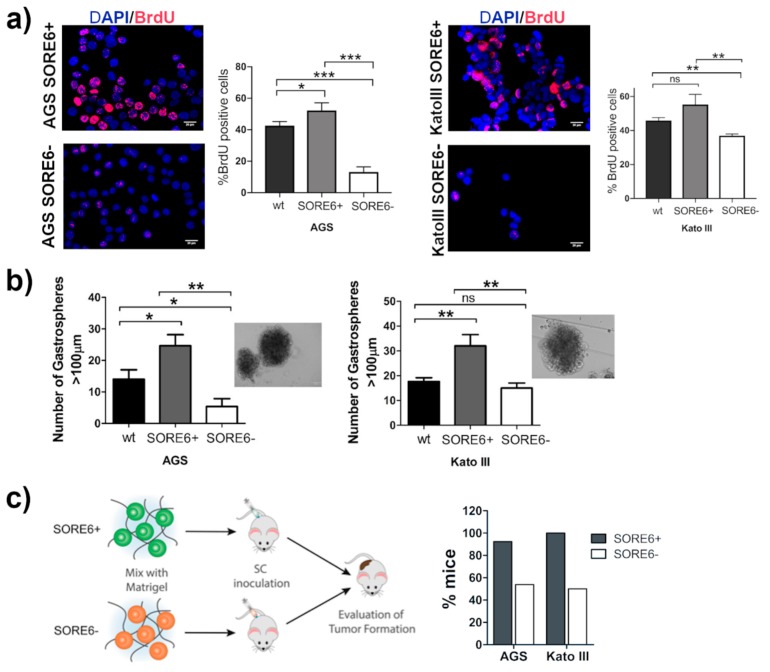
SORE6+ cells have a higher ability to proliferate, to form gastrospheres in non-adherent conditions and a larger in vivo tumor-initiating capability than SORE6− cells. (**a**) Immunofluorescence displaying the BrdU incorporation in AGS and Kato III SORE6 subpopulations followed by the results obtained by FACS analysis of BrdU incorporation in AGS wt, SORE6+, and SORE6− cells and Kato III wt, SORE6+, and SORE6− cells to evaluate the proliferation ability of each cell line (Appendix A). Scale bar = 100 µm. (**b**) Gastrosphere-forming ability of AGS wt, SORE6+, and SORE6− cells and Kato III wt, SORE6+, and SORE6− cells. Only spheres with a diameter >100 µm were considered as gastrospheres. Representative phase-contrast images of the gastrospheres are shown. Scale bar = 100 µm. Results are mean ± SD of three independent experiments. Significant differences (* *p* ≤ 0.05; ** *p* ≤ 0.01 and *** *p* ≤ 0.001). ns: Not significant. (**c**) Percentage of mice that developed a tumor after subcutaneous inoculation of 5 × 10^5^ AGS SORE6+ or SORE6− cells or with 3 × 10^5^ Kato III SORE6+ or SORE6− cells.

**Figure 4 cancers-12-00495-f004:**
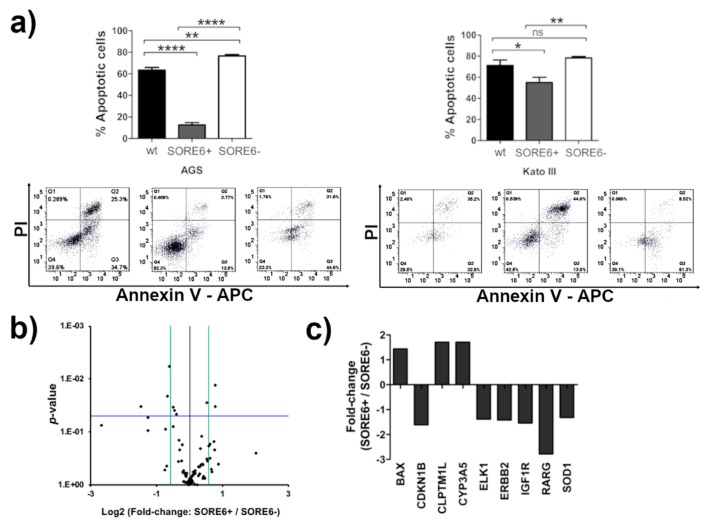
SORE6+ cells are more resistant to 5-fluorouracil (5-FU) than SORE6− cells. (**a**) Annexin V/ propidium iodide (PI) assay results after FACS analysis of AGS and Kato III wt and SORE6 subpopulation after 48 h treatment with 5-FU. Results are mean ± SD of three independent experiments. Significant differences (* *p* ≤ 0.05; ** *p* ≤ 0.01, and **** *p* ≤ 0.0001), ns-no significant. (**b**) Gene expression analysis of 84 genes involved in the response to chemotherapy in AGS SORE6+ and SORE6− cells using the RT2 Profiler PCR Array Human Cancer Drug Resistance. Volcano plot of AGS SORE6+ cells in comparison to AGS SORE6− cells (*n* = 3). The horizontal blue line represents the threshold of statistical significance (*p* = 0.05) and the green lines corresponds to the fold change cut-off ≥1.5. (**c**) Genes that showed a significant fold change, up- or down-regulation, (*p* ≤ 0.05) in AGS SORE6+ cells compared to AGS SORE6− cells.

**Figure 5 cancers-12-00495-f005:**
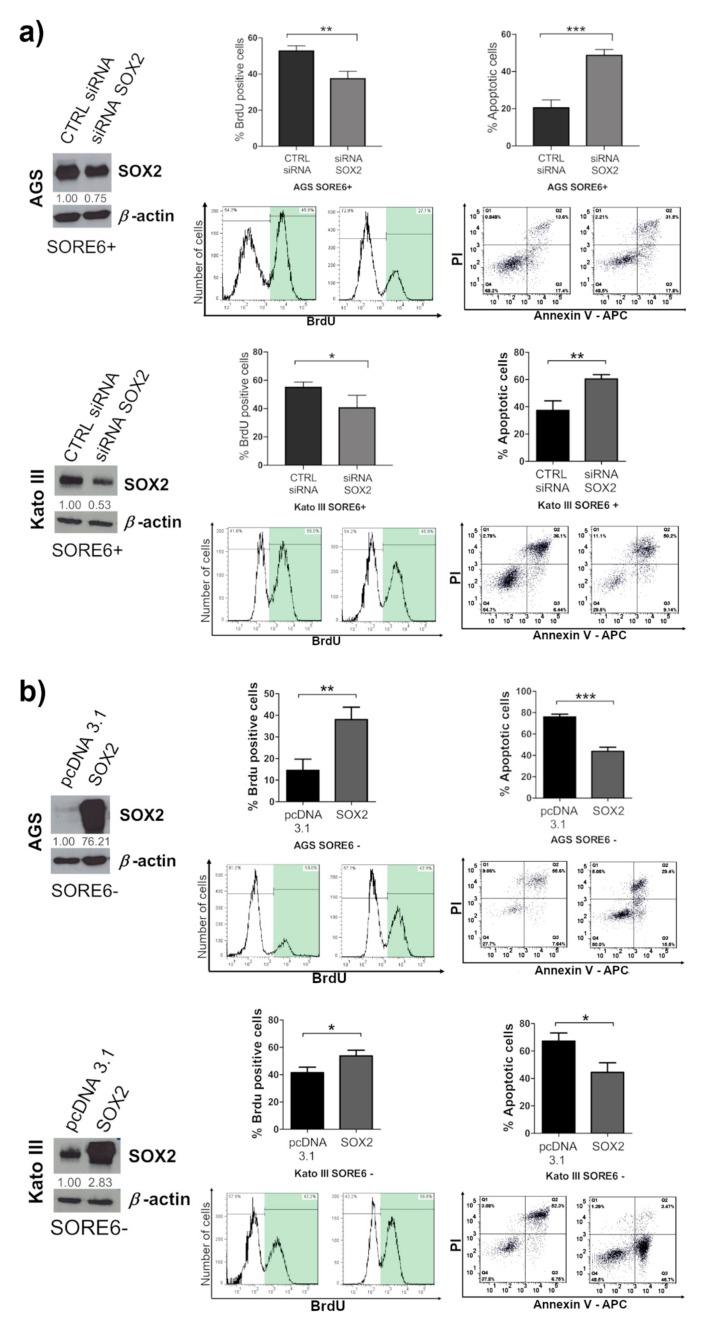
SOX2 has a key role on cell proliferation and on 5-FU resistance. (**a**) Western blot of AGS SORE6+ and Kato III SORE6+ cells 72 h after transient transfection with an siRNA for knockdown of the SOX2 protein (siRNA SOX2); the corresponding control is CTRL siRNA. This is followed by the results of BrdU incorporation for proliferation analysis and annexin V/PI assay results. Cells were used for apoptosis analysis after 48 h incubation with 5-FU. (**b**) Western blot of AGS SORE6− and Kato III SORE6− cells 72 h after transient transfection with the aim of overexpressing the SOX2 protein (SOX2); the corresponding control is pcDNA3.1. This is followed by the results of BrdU incorporation for proliferation analysis and annexin V/PI assay results, used for apoptosis analysis after 48 h of incubation with 5-FU. In Western blot *β*-actin was used as an internal control. Numbers below every Western blot picture represent semi-quantitative analysis of each line (intensity ratio: SOX2/*β*-actin). Pictures show cropped areas of Western blots, the whole images are included in the Appendix A. Results are the mean ± SD of three independent experiments. Significant differences (* *p* ≤ 0.05, ** *p* ≤ 0.01, and *** *p* ≤ 0.001).

**Figure 6 cancers-12-00495-f006:**
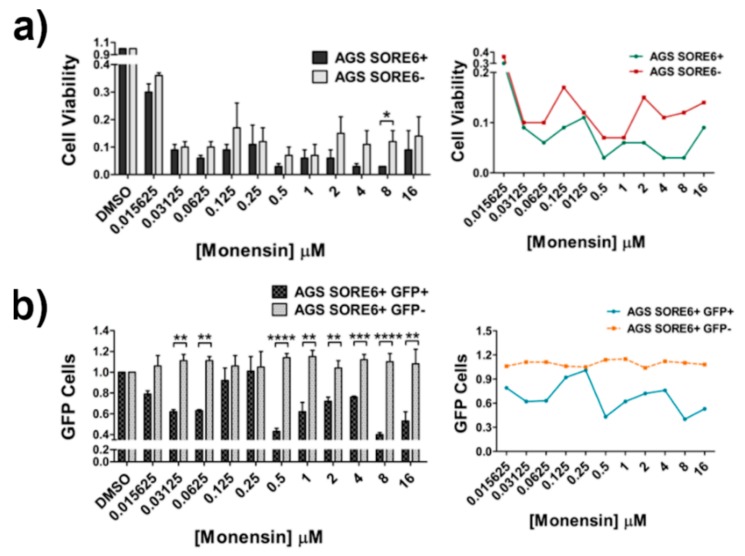
Monensin is more effective in SORE6+ cells. (**a**) AGS SORE6+ and SORE6− cells viability results after 48 h of treatment with monensin, normalized for the DMSO. (**b**) Number of GFP+ and GFP− cells in the AGS SORE6+ population after 48 h of treatment with monensin, normalized for the DMSO. Results are mean ± SD of three independent experiments. Significant differences (* *p* ≤ 0.05, ** *p* ≤ 0.01, *** *p* ≤ 0.001, and **** *p* ≤ 0.0001).

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
