# Peer review of "A SOX2 Reporter System Identifies Gastric Cancer Stem-Like Cells Sensitive to Monensin"

_cancers, 2020, doi:10.3390/cancers12020495_

Round 1

Reviewer 1 Report

Pádua et al demonstrated a platform technology for development of therapeutics for targeting gastric cancers using SOX2/OCT4 reporter assay. They isolated SORE6-GFP-positive gastric cells as SOX2-expressing CSC and characterized them. Moreover, they identified monensin as a SORE6+-killing agent. Overall strategy and some results are interesting. However, several points should be clarified with further experiments.

Fig. 2. Authors isolated SORE6-GFP-positive gastric cells and performed western blot analysis of biomarker of CSC. However, they did not examined expression level of SOX2 target genes, such as c-MYC, WNT1, WNT2 or NOTCH1. Because they isolated SOX2-expressing cells, it is necessary to observe expression levels of SOX2 target genes by western blot.

Fig3. a) Authors evaluated proliferation rate of SORE6+ cells by BrdU assay using FACS Aria II. I suggest authors to perform BrdU assay for imaging and show the incorporated BrdU in the proliferating cells by microscopic observation. c) Authors performed xenograft assay using SORE6+ and SORE6- cells. They only mentioned % of mice with tumor formation. They should show average size and characteristics of tumor (H&E or SOX2 IHC) derived from SORE6+ and SORE6- cells.

Fig. 4. a) They demonstrated SORE6+ cells are more resistant to 5FU than SORE6- cells. They counted apoptotic cells by annexin V/PI assay. I would like to ask authors to compare degree of apoptosis by visualizing caspase activity or western blots of apoptotic-specific proteins such as PARP, caspase 3 or caspase 8 in SORE6+ and SORE6- cells. c) To understand molecular mechanism in drug resistance in SORE6+ cells, authors screened mRNAs showing difference in their expression levels in SORE6+ cells in PCR Array. Authors should confirm mRNA or protein levels of the genes identified in PCR array in both SORE6+ and SORE6- cells.  

Fig. 5. For proliferation and apoptosis assay, author need to show clear visualized results in addition to bar graphs as described above.

Fig. 6. Cytotoxic drugs show dose-dependent growth inhibition of cancer cells. Cell viability of SORE6+ cells did not show dose dependency in the presence of monensin. Please explain this result. How many times of experiments are repeated?  

Some paper showed that SOX2 is downregulated in GC and inhibits cell growth. Please mention it in the “Introduction” or “Discussion” section.

Author Response

Pádua et al demonstrated a platform technology for development of therapeutics for targeting gastric cancers using SOX2/OCT4 reporter assay. They isolated SORE6-GFP-positive gastric cells as SOX2-expressing CSC and characterized them. Moreover, they identified monensin as a SORE6+-killing agent. Overall strategy and some results are interesting. However, several points should be clarified with further experiments.

Fig. 2. Authors isolated SORE6-GFP-positive gastric cells and performed western blot analysis of biomarker of CSC. However, they did not examined expression level of SOX2 target genes, such as c-MYC, WNT1, WNT2 or NOTCH1. Because they isolated SOX2-expressing cells, it is necessary to observe expression levels of SOX2 target genes by western blot.

We thank the reviewer for this suggestion. We have now analysed the expression of two of the identified SOX2 targets, c-MYC and NOTCH1, and observed increased expression of both in the SORE6+ subpopulation, in both cell lines. We have included these results in the revised version of the Ms as Supplementary Figure 1A. We changed the RESULTS and MATERIAL and METHODS sections accordingly, and included a new reference to support this study (ref 48).

Fig3. a) Authors evaluated proliferation rate of SORE6+ cells by BrdU assay using FACS Aria II. I suggest authors to perform BrdU assay for imaging and show the incorporated BrdU in the proliferating cells by microscopic observation.

According to the Reviewer’s suggestion we have now included, in the revised version of the Ms, analysis of BrDU incorporation performed by immunofluorescence. This result was added to Figure 3 and Supplementary Figure 2A. MATERIAL and METHODS was changed accordingly.

c) Authors performed xenograft assay using SORE6+ and SORE6- cells. They only mentioned % of mice with tumor formation. They should show average size and characteristics of tumor (H&E or SOX2 IHC) derived from SORE6+ and SORE6- cells.

We observed that the size of the tumors were very heterogeneous, even within the same cell subpopulation, and considered that it might not represent the tumorigenic capacity of the SORE6+ and SORE6- cells, for two reasons: 1. Cells were injected together with Matrigel, which might introduce a bias in the size of the tumors, at least in the early stages; 2. Tumors exhibited variable extensions of necrosis, which might change their size (and explain the heterogeneity observed) irrespective of the tumorigenic capacity of the cells. We have now included a picture representative of the tumors obtained with AGS SORE6+ cells and AGS SORE6- cells (Supplementary Figure 2C) to highlight the variable sizes.

Regarding H&E and SOX2 IHC of the xenografts we have now included a panel showing these results in tumors obtained from SORE6+ and SORE6- cells from both cell lines. We observed that tumors obtained from SORE6+ cells expressed SOX2 abundantly, as expected. We also observed some expression of SOX2 in tumors obtained from the SORE6- subpopulations. This was expected in KATOIII cell line, as SOX2 was never completely absent in the SORE6- cells. In contrast, in AGS SORE6- cells, SOX2 expression was completely absent in vitro but regained to some extent in vivo, suggesting phenotypic plasticity as previously described by Tang et al (ref 47). This result was included in Supplementary Figure 2D and described in the RESULTS section. MATERIAL and METHODS was changed accordingly.

Fig. 4. a) They demonstrated SORE6+ cells are more resistant to 5FU than SORE6- cells. They counted apoptotic cells by annexin V/PI assay. I would like to ask authors to compare degree of apoptosis by visualizing caspase activity or western blots of apoptotic-specific proteins such as PARP, caspase 3 or caspase 8 in SORE6+ and SORE6- cells.

According to the Reviewer’s suggestion we have now included western blots for cleaved PARP and caspases/cleaved caspases 3, 7 and 9. The results obtained corroborate the previous results observed with Annexin V. They show that the basal level of apoptosis is higher in KatoIII than in AGS but it increases in the presence of 5FU, particularly in the SORE6- subpopulations. The results obtained also show that apoptosis is clearly caspase-dependent in AGS but not in KatoIII, as previously described (ref 50). We have included this result in Supplementary Figure 3A, modified the RESULTS and MATERIAL and METHODS accordingly and included a new reference (ref 50).

c) To understand molecular mechanism in drug resistance in SORE6+ cells, authors screened mRNAs showing difference in their expression levels in SORE6+ cells in PCR Array. Authors should confirm mRNA or protein levels of the genes identified in PCR array in both SORE6+ and SORE6- cells.

According to the Reviewer’s suggestion we have now compared the expression levels of 6 differentially expressed genes, between SORE6+ and SORE6- cells, in AGS and KatoIII, by qRT-PCR. We studied the expression of the three genes that were up-regulated in AGS SORE6+ cells: BAX, CLPTM1L and CYP3A5 and of three that were downregulated: CDKN1B, RARG and SOD1. The results obtained corroborated the screening in AGS, whereas in KatoIII only the upregulated genes were confirmed, possibly indicating the most relevant mechanisms in the context of CSCs and suggesting that there are cell type specific mechanisms. We have now included this result in Supplementary Figure 3B and modified the RESULTS and MATERIAL and METHODS accordingly.

Fig. 5. For proliferation and apoptosis assay, author need to show clear visualized results in addition to bar graphs as described above.

We have included in the revised version of the Ms the flow cytometry results, analysing proliferation and apoptosis, in Figure 4, Figure 5 and Supplementary Figure 2. Legends to figures were modified accordingly.

Fig. 6. Cytotoxic drugs show dose-dependent growth inhibition of cancer cells. Cell viability of SORE6+ cells did not show dose dependency in the presence of monensin. Please explain this result. How many times of experiments are repeated?

This experiment was repeated 3 times, in triplicates, for each cell line. As the Reviewer pointed out, the effect of monensin is not completely dose-dependent, possibly due to the high toxicity of the compound in concentrations above 0.016 uM, particularly in AGS cells. In KatoIII, that are less sensitive to the compound, a dose-dependent effect was observed. However, we used high concentrations since these are the ones that have an impact in the number of GFP+ cells, within the SORE6+ subpopulations.

Some paper showed that SOX2 is downregulated in GC and inhibits cell growth. Please mention it in the “Introduction” or “Discussion” section.

We have introduced this sentence in the Discussion, of the revised version of the Ms, and cited a new reference (ref 56).

Reviewer 2 Report

The manuscript is quite well written.

The authors demonstrated that SORE6-GFP is a novel reporter system to recognize cancer stem-like cells facilitates our understanding of gastric CSCs biology and serve as a platform for the identification of powerful therapeutics for targeting gastric CSCs.

The methods are adequate.

The results justify the conclusions drawn.

It would be useful for the readers to include the discussion ofPMID: 31822580.

Author Response

The manuscript is quite well written.

The authors demonstrated that SORE6-GFP is a novel reporter system to recognize cancer stem-like cells facilitates our understanding of gastric CSCs biology and serve as a platform for the identification of powerful therapeutics for targeting gastric CSCs.

The methods are adequate.

The results justify the conclusions drawn.

We thank the Reviewer for the positive comments.

It would be useful for the readers to include the discussion of PMID: 31822580.

We have added this reference to the DISCUSSION of the revised version of the Ms (ref. 61).

Round 2

Reviewer 1 Report

Authors revised the mauscript according to questions raised. However, they did not include discussion the following question. I list two papers here.  They can refer them.

Some paper showed that SOX2 is downregulated in GC and inhibits cell growth. Please mention it in the
“Introduction” or “Discussion” section.

T Otsubo et al., SOX2 is frequently downregulated in gastric cancers and inhibits cell growth through cell-cycle arrest and apoptosis British Journal of Cancer 98, pages824–831(2008) Carrasco-Garcia et al., Paradoxical role of SOX2 in gastric cancer Am J Cancer Res 2016;6(4):701-713

Author Response

Authors revised the mauscript according to questions raised. However, they did not include discussion the following question. I list two papers here.  They can refer them.

Some paper showed that SOX2 is downregulated in GC and inhibits cell growth. Please mention it in the

“Introduction” or “Discussion” section.

T Otsubo et al., SOX2 is frequently downregulated in gastric cancers and inhibits cell growth through cell-cycle arrest and apoptosis British Journal of Cancer 98, pages824–831(2008) Carrasco-Garcia et al., Paradoxical role of SOX2 in gastric cancer Am J Cancer Res 2016;6(4):701-713

We apologize for the poor understanding of this point. The Reviewer raises an important issue concerning the conflicting data about SOX2 role in gastric cancer. We and others have shown that SOX2 associates with aggressive features, poor patient outcome and worse response to chemotherapy. However, other studies have shown the contrary and suggest a tumor suppressive role for SOX2 in gastric cancer. This is now clearly described in INTRODUCTION and supported with two new references, as suggested by the Reviewer.